# Transgene-Free Cynomolgus Monkey iPSCs Generated under Chemically Defined Conditions

**DOI:** 10.3390/cells13060558

**Published:** 2024-03-21

**Authors:** Yuliia Tereshchenko, Nesil Esiyok, Enrique Garea-Rodríguez, Daniele Repetto, Rüdiger Behr, Ignacio Rodríguez-Polo

**Affiliations:** 1Research Platform Degenerative Diseases, German Primate Center-Leibniz Institute for Primate Research, Kellnerweg 4, 37077 Göttingen, Germany; ytereshchenko@dpz.eu (Y.T.); nesiyok@dpz.eu (N.E.); 2German Center for Cardiovascular Research (DZHK), Partner Site Göttingen, 37075 Göttingen, Germany; 3Charles River Laboratories Germany GmbH, Hans-Adolf-Krebs-Weg 9, 37077 Göttingen, Germany; enrique@apollo.vc (E.G.-R.); daniele.repetto@crl.com (D.R.); 4Department of Developmental Biology, Göttingen Center for Molecular Biosciences, University of Göttingen, Justus-von-Liebig Weg 11, 37077 Göttingen, Germany

**Keywords:** macaque, iPSC, stem cell, non-human primate, neuronal differentiation, cardiac differentiation, regeneration

## Abstract

Non-human primates (NHPs) are pivotal animal models for translating novel cell replacement therapies into clinical applications, including validating the safety and efficacy of induced pluripotent stem cell (iPSC)-derived products. Preclinical development and the testing of cell-based therapies ideally comprise xenogeneic (human stem cells into NHPs) and allogenic (NHP stem cells into NHPs) transplantation studies. For the allogeneic approach, it is necessary to generate NHP-iPSCs with generally equivalent quality to the human counterparts that will be used later on in patients. Here, we report the generation and characterization of transgene- and feeder-free cynomolgus monkey (*Macaca fascicularis*) iPSCs (Cyno-iPSCs). These novel cell lines have been generated according to a previously developed protocol for the generation of rhesus macaque, baboon, and human iPSC lines. Beyond their generation, we demonstrate the potential of the novel Cyno-iPSCs to differentiate into two clinically relevant cell types, i.e., cardiomyocytes and neurons. Overall, we provide a resource of novel iPSCs from the most frequently used NHP species in the regulatory testing of biologics and classical pharmaceutics to expand our panel of iPSC lines from NHP species with high relevance in preclinical testing and translational research.

## 1. Introduction

Due to their origin, in vitro expansion potential, and capacity to differentiate into any cell of the adult body, induced pluripotent stem cells (iPSCs) hold great potential for the treatment of degenerative diseases. The generation of advanced therapy medicinal products (ATMPs) derived from iPSCs has been the focus of study of many research groups since the development of iPSC technologies in 2006 [1,2]. Nowadays, using iPSCs, it is possible to generate a large number of different functional cell types in vitro, e.g., neurons and cardiomyocytes, which can be transplanted into patients. Many reports have been published showing promising results regarding the treatment of diseases like myocardial infarction, Parkinson’s disease, or macular degeneration [3]. Over the last few years, the field has broadly diversified, and, considering the diverse upcoming novel therapeutic strategies for the treatment of diverse diseases, it is necessary to adapt and optimize the panel of translational animal models that allow the efficient translation of these novel therapeutic approaches to the clinics.

For the preclinical testing of regenerative therapies, research mainly relies on large animal models such as pigs and non-human primates (NHPs), depending on the specific questions each study aims to address [4,5,6,7]. In particular, NHPs emerged as invaluable models to study the potential of stem cell-based regenerative therapies, due to their similarities in physiology, immunology, and genetics to humans [6,7].

To generate robust preclinical safety and efficacy data for ATMPs, ideally, both xenogenic and autologous/allogenic transplantation strategies of ATMPs should be tested. In xenogenic transplantation studies in NHPs, human iPSC-derived tissue-specific cells are usually transplanted into the NHP. These studies mainly address the characteristics and the safety (tumorigenicity) of the actual ATMPs that will be used later in patients. In autologous/allogenic transplantation studies, species-specific iPSC-derived ATMPs are transplanted into the NHP. This approach addresses survival, immunological aspects, safety, and functionality [7].

To perform translational allogenic studies, it is expedient to generate NHP-iPSCs of quality and developmental potential that are homologous to their human counterparts. However, NHP-PSC technologies have lagged behind human or mouse stem cell technologies [7]. Nevertheless, several reports over the last few years have expanded the panel of NHP-iPSCs generated under feeder-free and transgene-free conditions suitable for preclinical testing [8,9,10]. This expanding panel of species includes the baboon, rhesus macaque, cynomolgus macaque, and marmoset [7,11].

In a previous study [9], we reported a robust protocol to generate transgene- and feeder-free iPSCs from human, rhesus macaque, and baboon cells using episomes and chemically defined cell culture conditions. These cells have demonstrated their usefulness not only for the validation of cell replacement therapies but are also currently used in several projects addressing primate developmental and evolutionary questions. Therefore, in the current work, we have successfully applied our previously published universal primate reprogramming and maintenance conditions to cynomolgus macaque cells (*Macaca fascicularis*). This is of major relevance as this NHP species is broadly and predominantly used in pharmaceutical research, development, and toxicity testing [12,13,14,15]. We have generated and characterized four novel cynomolgus monkey iPSC lines, the differentiation capacities of which were exemplarily tested by directed differentiation into two clinically relevant cell types, i.e., neurons and cardiomyocytes. The cell lines generated from the best-established NHP species in regulatory safety testing represent a significant expansion of the panel of NHP-iPSC lines available for the translation of preclinical findings to the clinics.

## 2. Materials and Methods

### 2.1. General Statement Addressing the Methods Used in This Study

This communication complements our previous study [9], expanding the versatility of the protocol described previously regarding an additional NHP species with major relevance in regulatory testing. Therefore, the methods for the generation, keeping, and differentiation of the Cyno-iPSCs are the same as those previously published. Detailed descriptions of the methods have been published recently [9,16].

### 2.2. Animals

Samples from two adult healthy and untreated male cynomolgus macaques (*Macaca fascicularis*) were used for fibroblast isolation. The tissue samples were collected in accordance with §4 (3) of the German Animal Welfare Act (killing for scientific purposes) at Charles River Laboratories during necropsy. The primary purpose was the removal of monkey hearts, lungs, and testes, which was approved by the responsible animal welfare officer. The treatment of the animals and their euthanasia was carried out in accordance with all legal guidelines and requirements.

### 2.3. Isolation and Cultivation of Macaque Primary Fibroblasts

Biopsies from the skin, gingiva, and foreskin were collected in ice-cold PBS (phosphate-buffered saline) with antibiotics (1%, *v*/*v*, penicillin/streptomycin, Gibco, Grand Island, NY, USA). Fibroblasts were extracted from tissue according to the method used in [17]. In brief, the tissue was minced using scissors and scalpels and digested in Dulbecco’s modified Eagle’s medium (DMEM) (Gibco), supplemented with 10 mg mL^−1^ collagenase type IV (Gibco), for 1–2 h at 37 °C in a Falcon tube rotator at 40 rpm. After digestion, the cell suspension was centrifuged (300× *g*; 5 min; room temperature). After centrifugation, the supernatant was removed and the cell pellet resuspended in Rh15 medium (DMEM, Gibco, 15%, *v*/*v*; fetal bovine serum, Gibco, 1%, *v*/*v*; penicillin/streptomycin, Gibco, 0.25 µg mL^−1^; amphotericin B, Sigma-Aldrich (Taufkirchen, Germany), 1%, *v*/*v*; minimum essential medium (MEM) non-essential amino acids solution, Gibco; 2 mM GlutaMAX, Gibco). Gingiva-derived fibroblasts were kept with a double amount of antibiotics (2%, *v*/*v*; penicillin/streptomycin, Gibco, 0.25 µg mL^−1^; amphotericin B, Sigma-Aldrich, 2%, *v*/*v*) for 4 days. Fibroblasts were expanded in Rh15 medium, using StemPro Accutase for dissociation (Thermo Fisher, Waltham, MA, USA).

### 2.4. Reprogramming NHP and Human Skin Fibroblasts and iPSC Maintenance

Reprogramming of the fibroblasts was performed by the nucleofection of primary fibroblast (6 × 10^5^ cells per transfection) with episomal vectors *pCXLE-hOCT3/4-shp53-F* (Addgene #27077), *pCXLE-hSK* (#27078) and *pCXLE-hUL* (#27080) (2 µg of each plasmid per transfection, 6 µg total plasmid DNA) [18] using the 4D-Nucleofector (P2 Primary Cell Solution) (Lonza, Basel, Switzerland). The transfected fibroblasts were cultured for 1 day in Rh15 medium supplemented with 10 ng/mL bFGF, 5 µM pro-survival compound (ROCKi; Merck, Darmstadt, Germany), and penicillin-streptomycin. On day 1, after transfection, the medium was supplemented with 10 ng/mL bFGF and 0.5 mM sodium butyrate (Sigma-Aldrich). After 7 days, the cells were transferred to Geltrex™ (0.16 mg/mL; Thermo Fisher)-coated tissue culture plates. On day 8, the Rh15 medium was substituted by Essential 8 (E8) medium (Thermo Fisher) supplemented with 0.5 mM sodium butyrate until day 11. From day 12, the cells were cultured in E8 medium only. After 30 to 40 days, the iPSC colonies were picked manually and transferred to freshly coated plates.

Putative iPSC lines were cultured in a universal primate pluripotent stem cell (UPPS) medium [9], StemMACS iPS Brew XF (Miltenyi Biotec, Bergisch Gladbach, Germany), with 1 µM inhibitor of Wnt response-1 (IWR-1, Sigma-Aldrich) and 0.5 µM of the GSK-3α/β inhibitor Chir99021 (Merck) and were then passaged in clumps using Versene solution (Thermo Fisher).

### 2.5. In Vitro Differentiation

Cyno-iPSCs were differentiated using the embryoid body (EB) formation assay [9,16]. In brief, cells were dissociated using collagenase type IV into cell clumps that were cultured in suspension in UPPS medium for 24 h. One day later, the medium was changed for differentiation medium (79 mL IMDM (Thermo Fisher), 1 mL 100× NEAA, 20 mL FBS, 450 µM 1-thioglycerol (Sigma-Aldrich)). On day 8, the EBs in suspension were plated onto plates containing gelatin-coated coverslips (0.1% gelatin) (Fisher Scientific, Schwerte, Germany) to allow them to attach. The attached EBs were maintained in culture until day 25, when they were fixed and analyzed by immunocytochemistry.

### 2.6. Cardiac Differentiation

The cardiac differentiation protocol has been adapted with minor modifications adopted from [19]. Cyno-iPSCs (iPSC#1.1, 1.2, and 2.1) were seeded in 12-well plates, with 90,000 cells/well, and cultured for two days. Then, the cells were exposed to mesodermal induction medium (RPMI 1640, B27 supplement without insulin (ThermoFisher), 200 µM L-ascorbic acid 2-phosphate (Sigma-Aldrich), 1 mM sodium pyruvate (ThermoFisher), 1 µM Chir99021 (Tocris, Bristol, UK), 9 ng/mL activin A (Miltenyi Biotec), and 5 ng/mL BMP4 (Miltenyi Biotec). The medium was changed after 24 h. On day 3, the medium was replaced by cardiac induction medium (RPMI 1640, B27 supplement without insulin, 1 mM sodium pyruvate, 200 µM L-ascorbic acid 2-phosphate, and 5 µM IWR-1). The medium was changed on day 5. On day 7, after the beginning of differentiation, the medium was changed to cardiomyocyte culture medium (RPMI 1640, B27 with insulin, and 200 µM L-ascorbic acid 2-phosphate). The cells were cultured further until day 11 and were then passaged using 1 mL/well TrypLE™ Express (ThermoFisher) for 10 min at 37 °C, centrifuged at 300× *g* for 5 min, and plated into Geltrex™-coated 6-well plates. After passaging, the iPSC-cardiomyocyte cultures were exposed for 7 days to selection by adding a culture medium containing lactate instead of glucose (RPMI 1640 without glucose (Thermo Fisher), 0.2 mg/mL L-ascorbic acid 2-phosphate, 4 mM lactate/HEPES solution, and 0.5 mg/mL recombinant human albumin) in order to further enrich the population in iPSC-cardiomyocytes [9]. Then, the medium was changed back to the cardiomyocyte culture medium and the cells were fed every second day for the next 7 days. Finally, the cells were processed for further analysis.

### 2.7. Neural Differentiation

The iPSCs (iPSC#1.1 and iPSC#2.1) were differentiated into neurons, generally based on the method of Qi et al. [4], following a protocol that was published recently [20]. In brief, iPSC cultures with 60–80% confluence were exposed for the first 7 days to neural induction medium (DMEM/F12 (Thermo Fisher Scientific), 10% KnockOut serum (KOS; Thermo Fisher Scientific), 1% non-essential amino acids (NEAA, Thermo Fisher Scientific), 200 µM L-ascorbic acid (L-AA, Sigma-Aldrich), 2 µM SB431542 (Peprotech, Hamburg, Germany), 3 µM Chir99021 (Sigma-Aldrich), and 1.5 µM dorsomorphin (Peprotech)). After induction, the cells were passaged (0.25% Trypsin/EDTA (Thermo Fisher Scientific)) and transferred to Geltrex-coated plates. At day 7, the medium was changed to neuralization medium containing DMEM/F12, 100 µM L-AA, 1% NEAA, 1× N2 supplement (Thermo Fisher Scientific), 1× B27 supplement (Thermo Fisher Scientific), 10 ng/mL bFGF (Peprotech), and 10 ng/mL EGF (Peprotech). After the neuralization phase, the cells were passaged once on poly-L-Lysin/laminin-coated plates (1 µg/mL poly-L-Lysin in water (Sigma-Aldrich) for 30 min at 37 °C) (2 µg/mL laminin in PBS (Sigma-Aldrich; #11243217001) for 8 h at RT) and maintained for 7 days in neuronal differentiation medium I, (DMEM/F12, 100 µM L-AA, 1% NEAA, 1× N2 supplement, 1× B27 supplement, and 300 ng/mL cAMP (Peprotech). Finally, for the last 7 days, the cells were maintained in differentiation medium II (DMEM/F12, 100 µM L-AA, 1% NEAA, 1× N2 supplement, 1X B27 supplement, 300 ng/mL cAMP, 10 ng/mL BDNF (Peprotech), and 10 ng/mL NT-3 (Peprotech)).

### 2.8. Immunostaining

Cells were fixed with paraformaldehyde (4% (*w*/*v*) Merck) in PBS (Thermo Fisher) for 20 min at room temperature and subsequently washed at least three times with PBS. Then, the cells were incubated for 30 min in 1% bovine serum albumin (BSA; Thermo Fisher) supplemented with 0.1% Triton X-100 (Sigma-Aldrich). After blocking/permeabilization, the cells were incubated at 4 °C overnight with primary antibody solutions (Table 1). Subsequently, secondary antibody (Table 1) incubation was performed for 1 h at 37 °C, followed by incubation with 1 µg/mL DAPI (4′,6-diamidino-2-phenylindole) (Sigma-Aldrich) for 5 min at room temperature. Finally, the coverslips were mounted using Fluoromount–G (Thermo Fisher) and imaged with an epifluorescence microscope (Zeiss, Oberkochen, Germany).

### 2.9. Flow Cytometry

Eleven days after the beginning of differentiation, the Cyno-iPSC-derived cardiomyocytes were dissociated with 1 mL/well TrypLE™ Express (ThermoFisher) for 10 min at 37 °C.

The single cells were fixed for 10 min at room temperature with 4% (*w*/*v*) paraformaldehyde. Subsequently, blocking/permeabilization was performed with 1% BSA supplemented with 0.1% Triton X-100 in PBS at 4 °C overnight. Afterwards, the cells were incubated with conjugated α-actinin (Vio^®^ R667) antibody solution (Table 1) at 37 °C for 1 h. Before analysis, the cells were washed with PBS and resuspended in 200 µL flow cytometry buffer (0.5% BSA, 2 mM EDTA (Carl Roth, Karlsruhe, Germany)). Flow cytometric analyses were performed with a SH800S Cell Sorter using a 100 µm chip (Sony Biotechnology, Weybridge, UK).

### 2.10. Nucleic Acid (DNA) Isolation and Polymerase Chain Reaction

Genomic DNA from the cell pellet samples was isolated using the DNeasy Blood & Tissue Kit (Qiagen, Hilden, Germany). The loss of episomal plasmids was demonstrated by a polymerase chain reaction (PCR) using plasmid-specific oligonucleotides (Sigma-Aldrich) (shown in Table 2) and polymerase with Standard Taq Buffer (New England Biolabs, Frankfurt, Germany). The different oligonucleotides and the PCR conditions were designed and validated for our previous study [9,16]. 

### 2.11. Protein Extraction and Western Blot Analysis

Protein from cultured cells (iPSCs) was isolated using RIPA lysis buffer (supplemented with Halt™ Protease Inhibitor Cocktail (100×), Thermo Fisher). The protein concentration in each sample was determined using the PierceTM BCA Protein Assay Kit (Thermo Fisher). For Western blot analysis, 10 µg of protein lysate (containing 1× DTT and 1× Laemmli sample buffer) was loaded onto a house-made 7.5% SDS-PAGE for protein separation. Following separation, the proteins were transferred onto an Immobilon^®^-P transfer membrane (Carl Roth, Karlsruhe, Germany). The membrane was then rinsed with TBS-T (1× TBS with 0.1% Tween-20) and blocked overnight at +4 °C using a solution of 5% powdered milk (Carl Roth) and TBS-T. Primary antibody incubation (Table 1) was carried out overnight at 4 °C. After washing with TBS-T, the membranes were incubated with a secondary HRP-conjugated antibody (Affinipure Goat Anti-rabbit IgG(H + L) and Anti-mouse (H + L) from Proteintech, no. SA00001-1 and SA00001-2). The Spectra™ Multicolor Broad Range Protein Ladder (Thermo Fisher) was used as a size reference. Signal detection was achieved using the chemoluminescent Pierce™ ECL Western Blotting Substrate (Thermo Fisher) and the ChemiDoc Western blot imaging system (Bio-Rad, Hercules, CA, USA).

## 3. Results

The aim of this study was to fine-tune the protocols to (1) generate iPSCs from an additional NHP species and (2) differentiate the iPSCs into clinically relevant cells under conditions that also work for human iPSCs [9]. We previously published a robust protocol to generate and characterize human, baboon, and rhesus macaque iPSCs. The present communication systematically tests the exact conditions in a different macaque species, the cynomolgus monkey, which represents the preferred NHP species in regulatory toxicity studies.

### 3.1. NHP and Human Fibroblast Reprogramming

We used tissue biopsies from two adult male macaques, hereinafter named Cyno#1 and Cyno#2, including the gingiva, foreskin, and skin. We successfully generated primary fibroblast lines from all tissues obtained from the two animals (exemplarily shown for Cyno#1 in Appendix A). For reprogramming, we selected an early passage of the fibroblast lines with a high proliferation rate and characteristic morphology. For one of the macaques, foreskin fibroblasts were selected (Cyno#1) for reprogramming, and for the other, the fibroblasts were derived from skin (Cyno#2). To induce pluripotency in the somatic cells, we used a set of episomal vectors for the transient expression of reprogramming factors [18]. Approximately 20 days after transfection, the first putative colonies became evident, and at day 30, colonies with a medium size were manually picked. As an early assessment of the pluripotency, alkaline phosphatase staining was performed in some of the primary plates on day 20. Alkaline phosphatase activity was detected in all cell clusters that showed primary colony morphology (Appendix A).

After the initial expansion of several putative iPSC lines until approximately passage five, two iPSC lines from each macaque were selected according to morphology for further passage, expansion, and characterization. The four iPSC lines (Cyno_iPSC#1.1 and Cyno_iPSC#1.2, from Cyno#1) (Cyno_iPSC#2.1 and Cyno_iPSC#2.2 from Cyno#2) showed the expected iPSC morphology, including compact colonies and sharp borders (Figure 1a). After ~15 passages, we evaluated the presence of the episomal vectors in the four iPSC lines by PCR. Two out of four cell lines showed no evidence of the transgenes at this passage (Figure 1b).

### 3.2. Cynomolgus Macaque iPSC Characterization

After derivation and expansion, the putative iPSC lines were characterized. Immunofluorescence analysis of the iPSCs revealed the expression of key pluripotency markers, including nuclear OCT4A, NANOG, SALL4, cytoplasmic LIN28, and the surface markers TRA-1-60 and TRA-1-81 (Figure 1c and Appendix A show nuclear marker colocalization with the nucleus). NANOG and SALL4 are not encoded by the reprogramming vectors. 

To further characterize the cell lines, we analyzed the developmental potency of the cells to confirm their pluripotent state using an embryoid body formation assay. The cells of the four cell lines aggregated under defined conditions (Figure 2). Additionally, after exposing the cell aggregates to a differentiation medium for more than 20 days, representative cell types of the three embryonic germ layers were detected in the EB outgrowths. Representative SMA (alpha-smooth muscle actin and mesoderm), beta III tubulin (ectoderm), and AFP (alpha-fetoprotein and endoderm)-positive cells were found in all cell lines (Figure 2).

### 3.3. NHP-iPSC-Derived Cardiomyocyte and Neurons

Different protocols are used to differentiate human iPSCs into tissue-specific cell types. However, one of the major difficulties in the field of preclinical testing is the translation of protocols of directed differentiation for human iPSCs to NHP-iPSCs. Previously, we have shown that rhesus macaque (*Macaca mulatta*) and baboon (*Papio anubis*) cells were refractory to directed differentiation following the protocols established for human cells [9]. After testing several approaches, we developed a protocol that was effective in human, rhesus macaque, and baboon cells. Here, we tested two different protocols in order to show (1) the translatability of those protocols to other macaque species and (2) further confirm the pluripotent state of the Cyno-iPSCs. As cardiovascular and neurodegenerative disorders have the highest clinical relevance, we decided to differentiate the Cyno-iPSCs exemplarily into neurons and cardiomyocytes. 

For the neuronal differentiation protocol, we used our recently published method [20] (Figure 3a). The differentiation protocol efficiently directed the cells to the neural lineage (tested for Cyno_1.1 and 2.1), and, after approximately 30 days of differentiation, the cell population was highly enriched in cells showing neuronal morphology, the downregulation of pluripotency factors (e.g., OCT4A), and expressed neuron-specific markers like beta-III-tubulin (Figure 3b,c).

For the cardiomyocyte differentiation protocol, we followed a protocol based on that used in [19] (Figure 3d). The Cyno-iPSCs (Cyno_iPSC#1.1, 1.2, and 2.1) differentiated into contracting cells after about ten days. After differentiation, the percentage of cardiomyocytes in the cell population was evaluated by analyzing the percentage of alpha actinin-positive cells by FACS. We observed significant heterogeneity in the efficiency of the different cell lines to differentiate into iPSC-CM, as shown for Cyno_1.1, Cyno_1.2, and Cyno_2.1, with 12%, 16%, and 58%, respectively (the gating strategy and individual values are shown in Appendix A). The differences in the propensity of the cell lines to differentiate into cardiomyocytes could be overcome after metabolic selection using lactate instead of glucose as the energy source. After 30 days in culture, the generated cardiomyocytes expressed cardiac-specific markers, as shown by staining and Western blot analysis for sarcomeric markers, and connexin 43, which is important for cell–cell communication between cardiomyocytes (Figure 3c,e,f).

## 4. Discussion

NHP are invaluable animal models to preclinically test cell replacement therapies. To generate predictive studies, it is necessary to count on a broad panel of iPSCs from the most relevant species in translational research. Old-world monkey species, like the rhesus macaque or baboon, are well established in this context; however, in the last few years, some key new-world monkey species, e.g., the marmoset, are gaining relevance. Additionally, to reinforce the translatability of such studies, the NHP-iPSCs need to meet the high standards of human iPSCs for generation and maintenance. Therefore, it is important to generate reprogramming protocols that are translatable between different primate species, including humans.

In this communication, we further expand the panel of primate species in which somatic cells can be reprogrammed following our previously established conditions using episomes and a chemically defined medium. Following this workflow, we report the generation of novel cynomolgus monkey iPSCs. This macaque species is of particular interest as it is the most widely used NHP species in regulatory testing and pharmaceutical research and development [12,13,14,15,21].

The generated novel iPSC lines present human iPSC-like morphology and cell behavior, including the potential to be expanded and maintained undifferentiated in UPPS medium for a high number of passages. Two of the novel cell lines are transgene-free, increasing their potential to be used by the scientific community and pharmaceutical industry in preclinical and regulatory studies in vivo and also in vitro, supporting and complementing in vivo studies. The four iPSC cell lines that have been characterized show the reactivation of the pluripotency network, as shown by the reactivation of pluripotency markers such as OCT4, NANOG, LIN28, SALL4, TRA-1-60, and TRA-1-81 revealed in the protein and transcript levels. 

Additionally, the novel iPSC lines’ differentiation potential was first evaluated by spontaneous differentiation by performing an embryoid body assay. Upon differentiation, the four different iPSC lines were able to generate representative cells from the three embryonic layers. Additionally, we validated two differentiation protocols to direct cynomolgus macaque iPSCs toward enriched populations of clinically relevant cell types, i.e., cardiomyocytes and neurons. The cardiomyocyte protocol that was previously validated for humans, baboons, and rhesus macaques can also robustly generate Cyno-iPSC-derived cardiomyocyte-like cells. These cells spontaneously contract and express cell type-specific markers. The variable efficiencies in differentiation can be overcome later on by the metabolic selection of the cultures, generating highly enriched cardiomyocyte-like cell populations (Figure 3d,f). The neuronal differentiation protocol, which was previously validated only in human iPSCs, can generate neuronal-like cells that present a characteristic morphology and express cell type-specific markers.

## 5. Conclusions

In this study, we generated novel iPSC lines from an NHP species with high biomedical relevance. Additionally, we evaluated the capacity of the novel iPSC lines to be differentiated towards the neural and cardiac fates, proving this cell’s capacity to be considered for the allogeneic transplantation of ATMP in preclinical studies using cynomolgus macaques.

## Figures and Tables

**Figure 1 cells-13-00558-f001:**
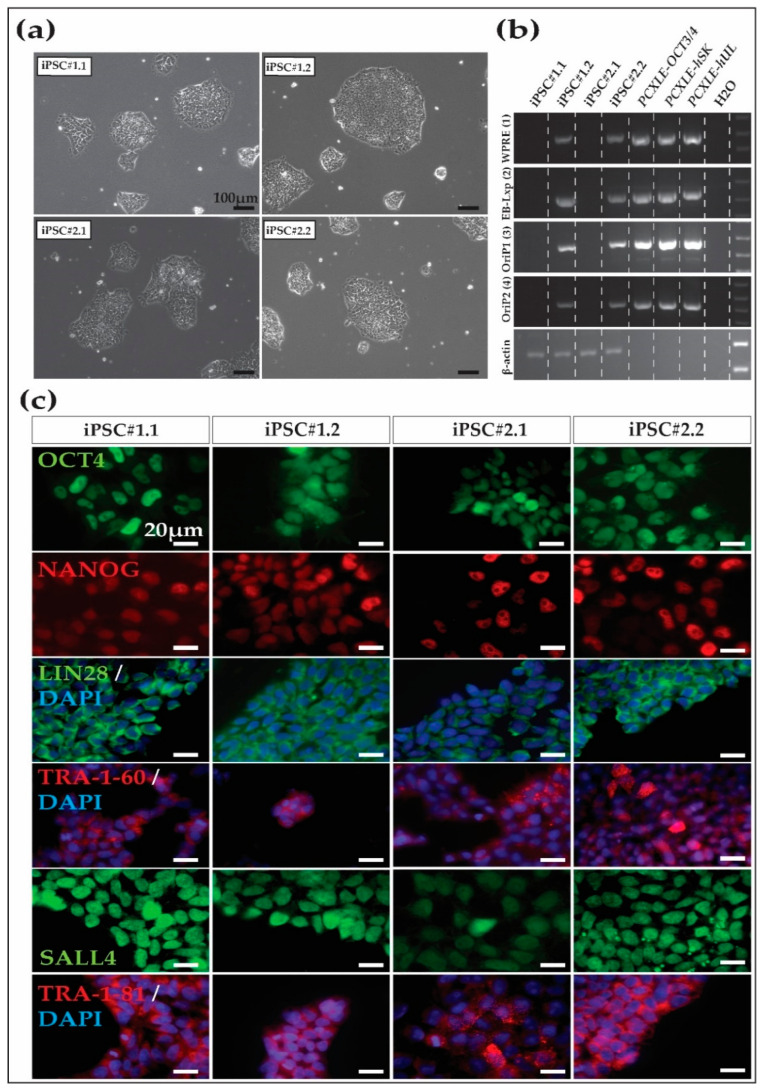
Cynomolgus macaque iPSC morphology and characterization. (**a**) Morphology of the four iPSC lines generated in this study and maintained in universal primate pluripotent stem cell (UPPS) medium. iPSC#1.1 and iPSC#1.2 were from Cyno#1, while iPSC#2.1 and iPSC#2.2 were from Cyno#2. Cynomolgus macaque iPSC colonies present human primed iPSC-like morphology, including clear borders and compact colonies consisting of cells that show a high nucleus-to-cytoplasm ratio. Scale bars 100 µm. (**b**) Two out of four iPSC cell lines are transgene-free at passage ~15, as shown by PCR using primers specific for different regions of the episomal reprogramming plasmids. (**c**) Pluripotency marker expression, as shown by immunofluorescence. Note that NANOG and SALL4 are not encoded by the reprogramming vectors. Scale bars 20 µm.

**Figure 2 cells-13-00558-f002:**
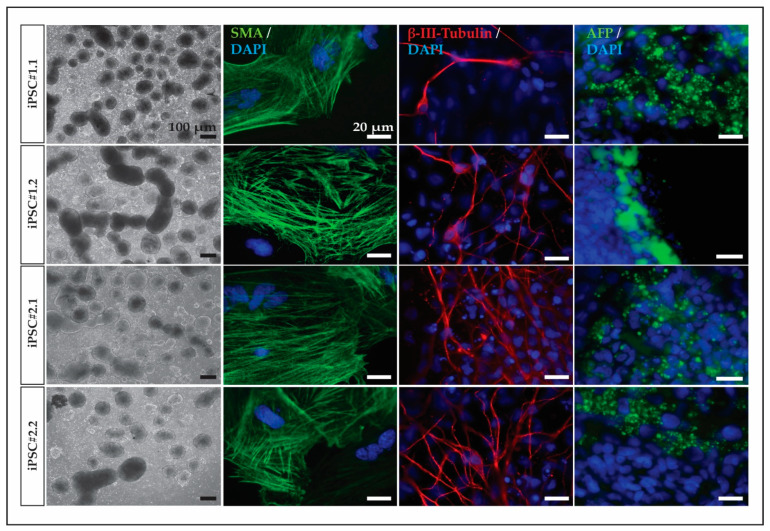
Embryoid body formation assay. The four macaque cell lines form cell aggregates and differentiate into representative cell types of the three embryonic germ layers. Staining was conducted for the markers β-tubulin III (ectoderm), alpha-smooth muscle actin (SMA, mesoderm), and α-fetoprotein (AFP, endoderm) in embryoid body outgrowths generated with DPZ_Cyno#1.1, DPZ_Cyno#1.2, DPZ_Cyno#2.1, and DPZ_Cyno#2.2. Scale bars 100 µm and 20 µm.

**Figure 3 cells-13-00558-f003:**
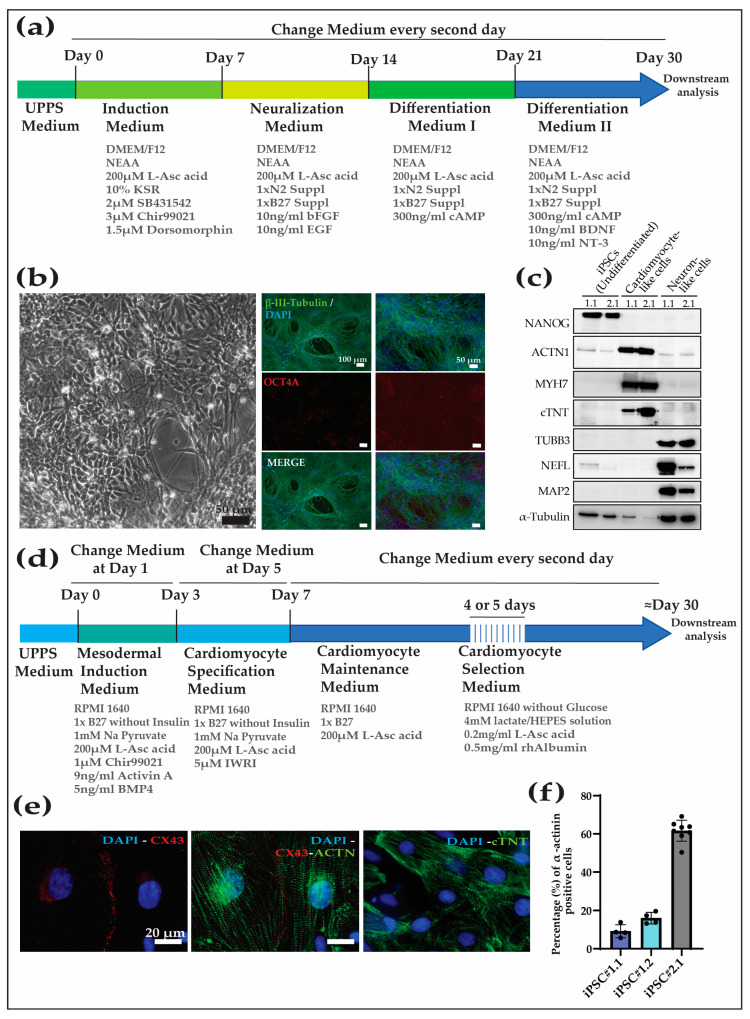
Neuron and cardiomyocyte differentiation protocols were applied to the Cyno-iPSCs. (**a**) Directed neuronal differentiation of Cyno-iPSCs. (**b**) Bright-field and immunofluorescence images of neuronal cells. The Cyno-iPSC-derived neurons expressed beta-III-tubulin and showed the characteristic morphology (shown for iPSC#2.1). (**c**) Protein abundance analysis was conducted on undifferentiated iPSCs and their differentiated counterparts, including cardiomyocyte- and neuron-like cells (Cyno_iPSC#1.1 and 2.1). The analysis encompassed one pluripotency marker (Nanog) and three markers specific to cardiomyocytes (ACTN1, MYH7, and cTNT), as well as three neuronal-specific markers (TUBB3, NEFL, and MAP2). Alfa-tubulin expression is used as a housekeeping marker. (**d**) Cardiac differentiation of Cyno-iPSCs. (**e**) Cyno-iPSC cardiomyocytes express cardiac-specific proteins, as shown by immunofluorescence staining (shown for Cyno_iPSC#2.1). The immunofluorescence of cardiac proteins shows the typical sarcomeric structures in the Cyno-iPSC-derived cardiomyocytes: sarcomeric α-actinin, cardiac troponin T (cTNT), and connexin 43 (Cx43). Scale bars, 20 µm. (**f**) Additionally, the efficiency of differentiation was evaluated by assessing the percentage of alfa-actinin-positive cells in the post-differentiation, pre-selection cell populations (Cyno_iPSC#1.1, #1.2, and #2.1) by fluorescence-activated cell sorting (FACS).

**Table 1 cells-13-00558-t001:** List of primary and secondary antibodies.

	Name	Company	Catalog #	Dilution
Primary Antibodies	α-feto protein	Dako (Glostrup, Denmark)	A0008	1:100
	α-actinin	Sigma-Aldrich	A7811	1:1000
	β-tubulin III	Sigma-Aldrich	T8660	1:1000
	Connexin 43 (Cx43)	Abcam (Cambridge, UK)	ab11370	1:1000
	α-actinin *	Miltenyi Biotec	130-128-591	1:50
	Cardiac troponin T (cTNT)	Miltenyi Biotec	130-106-687	1:10
	LIN28	R&D systems (Minneapolis, MN, USA)	AF3757	1:300
	NANOG	Cell Signalling (Danvers, MA, USA)	4903	1:400
	OCT4A	Cell Signalling	C52G3	1:1600
	TRA-1-81	Abcam	ab16289	1:200
	SALL4	Abcam	ab57577	1:200
	Smooth muscle actin	Sigma-Aldrich	A2547	1:1000
	TRA-1-60	Abcam	ab16288	1:200
	α-actinin #	Sigma-Aldrich	A7811	1:2000
	α-tubulin #	Cell Signalling	mAb#3873	1:5000
	MYH7 #	Sigma-Aldrich	HPA001239	1:1000
	Nanog #	Cell Signalling	D73G4	1:1000
	NEFL #	Sigma-Aldrich	N4142	1:2000
	MAP2 #	Sigma-Aldrich	HPA012828	1:2000
	B III tubulin #	Biolegend (San Diego, CA, USA)	Biolegend	1:2000
	cTNT #	Miltenyi Biotec	130-106-687	1:1000
Secondary antibodies	Alexa555-goat-α-mouse IgG	Thermo Fisher	A21424	1:1000
	Alexa488-goat-α-mouse IgG	Thermo Fisher	A11029	1:1000
	Alexa488-goat-α-mouse IgG/IgM	Thermo Fisher	A10680	1:1000
	Alexa488-donkey-α-goat IgG	Thermo Fisher	A11055	1:1000
	Alexa488-donkey-α-rabbit IgG	Thermo Fisher	A21206	1:1000

(*) Antibodies used for flow cytometry analysis; (#) antibodies used for Western blot analysis.

**Table 2 cells-13-00558-t002:** Oligonucleotides used in this study. The table includes the name, sequence, amplicon size (Amp) in base pairs, annealing temperature (T, in °C), and cycles used during DNA amplification. Primers used for the detection of the episomal plasmids (1–4), and β-actin primers used as positive control.

Name	Sequence	Amp	T	C
*WPRE* (1)	for: 5′-GCT ATT GCT TCC CGT ATG GC-3′	470	54	32
	rev: 5′-CAA AGG GAG ATC CGA CTC GT-3′			
*EBNA-LoxP* (2)	for: 5′-AAG AGG AGG GGT CCC GAG A-3′	555	61	32
	rev: 5′-GCC AAT GCA ACT TGG ACG TT-3′			
*OriP1* (3)	for: 5′-GGT TCA CTA CCC TCG TGG AAT-3′	592	57	32
	rev: 5′-CGG GGC AGT GCA TGT AAT-3′			
*OriP2* (4)	for: 5′-GGT GAC TGT GTG CAG CTT TG-3′	416	54	32
	rev: 5′-GGA GCT GAG TGA CGT GAC AA-3′			
*β-actin*	for: 5′-GAC CTG ACT GAC TAC CTC ATG-3′	380	61	32
	rev: 5′-GGT AGT TTC GTG GAT GCC ACA-3′			

## Data Availability

The cell lines generated in this study will be available upon request to the Research Platform Degenerative Diseases, German Primate Center-Leibniz Institute for Primate Research (to R.B.).

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
