# Peer review of "Transgene-Free Cynomolgus Monkey iPSCs Generated under Chemically Defined Conditions"

_cells, 2024, doi:10.3390/cells13060558_

Round 1

Reviewer 1 Report

Comments and Suggestions for Authors

In this short Communication, the authors described how they generated transgene-free Cynomolgus Monkey iPSCs. Thanks to already optimized protocols, they succeeded in obtaining iPSCs also from Macaca fascicularis which has been widely used in pharmaceutical research. In the work, the iPSCs status was well characterized by IF analysis showing the expression of the main staminal markers, then, the presence of episomal vectors was verify by PCR and the potential of the obtained lines to differentiate into embryonic germ layers was confirmed by IF. At the end of the work, preliminary tests to derive neuron and cardiomyocyte from the iPSCs were performed. I think that the workflow and the methods are clearly described and the aim of the work is well stated. Overall, the described results demonstrate the reaching of the goal stated in the abstract and in the introduction. Further, I think that this work could improve the knowledge of the NHP-derived iPSCs and can help in the development of new cellular-based therapies. However, I still have one major concern and I suggested few minor revisions that could implement the value of the proposed work.

Major concern

Although the IF and cytofluorimetric analysis could easily show the differentiation of IPSCs into neuron and cardiomyocyte, I think that a deeper investigation should be carried on confirming the differentiated state. I suggested PCR or Western Blot analysis for the specific markers. Then, the authors should justify the choice of #1.2 and #2.1 for cardiomyocyte differentiation and not the other lines. Further, in the text was not specified in which lines the neuronal differentiation was done. If the aim of the author is the demonstration of their capability to obtain differentiated cells for therapies or studies, they must better dissect this section.

Minor revision

In the paragraph: “Nucleic acid (DNA) Isolation and Polymerase Chain Reaction”, you have to add 2.10 to indicate the section. Then, I think could be helpful for the readers indicated how you designed or from who you bought the primers for PCR analysis.

In line 255 there is a typo. The authors described IF analysis, but they referred to Fig. 1b, which is the PCR results, and not to Fig. 1c.

If it is possible, I think that the authors may add in the supplementary materials the DAPI staining of sample marked for OCT4, NANOG and SALL4 to demonstrate their nuclear localization.

In paragraph 3.2, could you explained why the four iPSC lines showed different expression of AFP (#1.1 and #2.2 are similar and show higher expression, while #1.2 and #2.1 are less stained). If the problem is only in the selected pictures, I suggested to choose more representative ones.

Reviewer 2 Report

Comments and Suggestions for Authors

The article is devoted to obtaining integration-free iPSCs from primates. Such cells can be used to develop cell therapy techniques, which can then be translated into humans. The authors demonstrated the ability of the resulting iPSCs to effectively differentiate into neurons and cardiomyocytes. The manuscript was well planned and most of the experiments were carried out at a high methodological level. The abstract and introduction fully disclose the goals and objectives of the article.

However, there are a number of both minor and quite serious comments about the work:

1. Line 135 "were plated plates" - apparently the preposition is missing.

2. Table 2 should be placed after it is mentioned in the text, not before.

3. Line 255 - There is an error in the reference to the drawing, apparently it should be indicated Fig. 1С instead of FIg. 1b.

4. The AFP staining shown in Figure 2 is not what is expected for the antibodies used. For example, in the article doi: 10.1155/2018/3972353, where antibodies DAKO A0008 were also used, the staining pattern differs from that presented.

5. The difference in the efficiency of differentiation into cardiomyocytes between iPSC lines can be explained by the fact that in the iPSC1.2 line integration of reprogramming plasmids into the genome occurred (Fig. 1). Lines with potential integrations should be excluded from analysis because they do not meet the requirements for a cell therapy material.

Round 2

Reviewer 1 Report

Comments and Suggestions for Authors

Dear authors,

Thank you for addressing all my request. I think that the presented work should be published in this journal.

Reviewer 2 Report

Comments and Suggestions for Authors

I thank the authors for their detailed answers to all questions and comments. The work may be published in its present form.